# Facile Aqueous–Phase Synthesis of Pd–FePt Core–Shell Nanoparticles for Methanol Oxidation Reaction

Xiangyun Xiao [1,†], Euiyoung Jung [2,†], Sehyun Yu [3,†], Hyeonjin Kim [4,5], Hong-Kyu Kim [4], Kwan-Young Lee [5], Jae-Pyoung Ahn [4,*], Taeho Lim [3,*], Jinheung Kim [2,*] and Taekyung Yu [1,6,*]

1 Department of Chemical Engineering, Kyung Hee University, Yongin 17104, Korea; xyxiao@khu.ac.kr
2 Nanobio Energy Materials Center, Department of Chemistry and Nano Science, Ewha Womans University, Seoul 03760, Korea; jey9207@ewha.ac.kr
3 Department of Chemical Engineering, Soongsil University, Seoul 06978, Korea; sehyun7720@daum.net
4 Advanced Analysis Center, Korea Institute of Science and Technology, Seoul 02792, Korea; hjkim93@kist.re.kr (H.K.); hkkim@kist.re.kr (H.-K.K.)
5 Department of Chemical and Biological Engineering, Korea University, Seoul 02841, Korea; kylee@korea.ac.kr
6 Department of Chemical Engineering (BK21 FOUR Integrated Engineering Program), Kyung Hee University, Yongin 17104, Korea
* Correspondence: jpahn@kist.re.kr (J.-P.A.); taeholim@ssu.ac.kr (T.L.); jinheung@ewha.ac.kr (J.K.); tkyu@khu.ac.kr (T.Y.)
† These authors contributed equally to this work.

**Abstract:** Multi-metallic Pd@FePt core–shell nanoparticles were synthesized using a direct seed-mediated growth method, consisting of facile and mild procedures, to increase yield. The Fe/Pt ratio in the shell was easily controlled by adjusting the amount of Fe and Pt precursors. Furthermore, compared with commercial Pt/C catalysts, Pd@FePt nanoparticles exhibited excellent activity and stability toward the methanol oxidation reaction (MOR), making them efficient in direct methanol fuel cells (DMFC).

**Keywords:** multi-metal; core–shell; nanoparticles; direct seed-mediated growth; methanol oxidation reaction

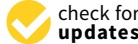



## 1. Introduction

Typically, noble metal nanoparticles have shown excellent performances in various applications such as sensing, biomedicine, and catalysis [1–3]. Moreover, to improve performance, many researchers have synthesized multi-component nanoparticles owing to the synergistic effects of different elements [4–6]. Xia et al. demonstrated that tri-metallic (Pt–Ir–Pd) nanoparticles as electrochemical catalysts showed superior activity than bi-metallic (Pt–Pd) nanoparticles toward oxygen reduction reaction and oxygen evolution reaction because of their electronic coupling effect [7]. However, the high cost and scarcity of noble metals can hinder their industrial use. Cost-effective materials and synthetic processes are often demanded to obtain catalysts with high activity and good stability. Thus, the incorporation of a transition metal can be a route to reducing the cost and increasing the catalytic performance [8,9]. Wheatley et al. reported that bi-metallic alloy (FePt) nanoparticles comprising transition metal oxide ($Fe_3O_4$) showed great oxygen reduction capability because of the donation of electron density from $Fe_3O_4$ to Pt [10]. FePt–$Fe_3O_4$ nanoparticles were synthesized using an organic solvent at a relatively high temperature (<200 °C) under argon atmosphere. These conditions can enable eco-friendly, low-cost synthetic procedures. We have recently developed a direct seed-mediated growth method for multi-metallic core–shell nanoparticles [11–14]. Compared with that in a typical seed-mediated growth method, the absence of washing and redispersion after seed synthesis can increase the efficiency by reducing the cost and time of nanomaterial production. By

the direct addition of metal precursors and by reducing agents to the reacting solution involving seeds, multi-metallic core–shell nanoparticles can be obtained.

Direct methanol fuel cells (DMFC) are promising candidates for portable devices because of their high energy density [15–17]. Methanol oxidation reaction (MOR) occurs at the anode of a DMFC, which is of increasing importance for practical applications. To date, Pt-based nanocatalysts have been commonly used; however, their activity needs to be developed using low-cost techniques.

In this study, we prepared multi-metallic core-shell (Pd@FePt) nanoparticles through the direct seed-mediated growth method under mild conditions. Specifically, we introduced an additional transition metal (Fe) precursor during shell formation with Pt on Pd seeds. We controlled the Fe/Pt atomic ratio in the shell by varying the amount of Fe and Pt precursors. Compared to commercial Pt/C catalyst, the carbon-supported Pd@FePt nanoparticles exhibited excellent activity and stability toward MOR.

## 2. Results

Pd@Fe and Pd@FePt core–shell nanoparticles were synthesized by a direct seed-mediated method in an aqueous solution. Fe and a mixture of Fe and Pt precursors for shell formation were used with $NaBH_4$ as a reducing agent, using Pd nanocubes as seeds. In an aqueous medium, the standard reduction potential of $NaBH_4$ was reported to be −1.33 V, which can reduce many metal cations such as copper, nickel, and Fe [18]. When we synthesized Pd@FePt nanoparticles, we used different molecular weight ratios of Fe and Pt precursors; that is, 3/1, 1/1, and 1/3, and nominally named them as $Pd@FePt_{3/1}$, $Pd@FePt_{1/1}$, and $Pd@FePt_{1/3}$, respectively. In inductively coupled plasma (ICP) results of the Pd@Fe and Pd@FePt nanoparticles, the atomic content of their shell was found to be 15–19%, which was well matched with the amount of injected Fe and Pt precursors (Table S1). Transmission electron microscopy (TEM) (Figure S1a–d) and high-resolution TEM (HRTEM) (Figure 1a–e) images showed that Pd@Fe, $Pd@FePt_{3/1}$, $Pd@FePt_{1/1}$, and $Pd@FePt_{1/3}$ nanoparticles had a cubic structure similar to Pd nanocubes. Figure S1e and S1f show the size distribution of Pd@Fe and Pd@FePt nanoparticles, indicating that their average particle size was around 13 nm. The X-ray diffraction (XRD) patterns of the nanoparticles showed the presence of intense peaks at $2\theta = 40.48°$, $48.72°$, $67.8°$, and $82.0°$, corresponding to (111), (200), (220), and (311) of metallic Pd with a face-centered cubic (FCC) crystalline structure, respectively (Figure 1f, JCPDS card no. 05-0681). The (111) peak from Pd@Fe nanoparticles shifted to a larger angle, possibly because of the lattice contraction caused by the presence of Fe having a smaller lattice than Pd. On the other hand, the diffraction peaks of Pd@FePt nanoparticles moved to a smaller angle with an increase in the amount of Pt in the FePt shell, indicating the formation of FePt alloy. The energy-dispersive X-ray spectroscopy (EDS) mapping images of Pd@Fe (Figure 2a) and Pd@FePt (Figure 2b–d) nanoparticles revealed that the Fe and FePt shells covered the surface of Pd nanocubes, respectively. Additionally, EDS line scanning images (Figure S2) confirmed that the Pd@Fe and Pd@FePt nanoparticles have a core–shell structure with Pd core and Fe or FePt alloy shell, respectively. Interestingly, Fe and FePt shells were grown on only the Pd core despite the large lattice mismatch between Pd and Fe without the formation of isolated nanoparticles. In a previous report on the synthesis of Pd@Pt core–shell nanocubes with controlled shell coverage, it was demonstrated that using a strong reducing agent could evenly deposit the shell on all facets of Pd cubes, thus leading to full-shell Pd@Pt core–shell nanocubes [19]. In this study, we believe that using a strong reducing agent led to the full-shell formation. When we used a weak reducing agent (L-ascorbic acid) instead of $NaBH_4$, $Pd@FePt_{1/3}$ had a concave structure with low Fe content (0.33 at%), which means that a strong reducing agent was essential to reduce transition metal ions (Figure S3). To obtain the chemical and electronic states of Fe and Pt on the Pd@Fe and Pd@FePt surfaces, X-ray photoelectron spectroscopy (XPS) was employed. XPS results showed that Fe 2p (Figure 3a and Figure S4), Fe $2p_{1/2}$, and $2p_{3/2}$ peaks of Pd@Fe nanoparticles are located at 724.2 eV and 710.8 eV, respectively, indicating that the Fe shell on the Pd@Fe and Pd@FePt

nanoparticles had an $Fe_3O_4$ structure [20]. The atomic ratio of Fe(III) to Fe(II) was about 2:1, which is similar to that in $Fe_3O_4$. It is worth noting that the EDS line scanning showed that the atomic ratio of O/Fe was about 4/3, implying the presence of the $Fe_3O_4$ structure (Figure S2). In Figure 3b, the Pt 4f spectrum of Pd@FePt nanoparticles shows the presence of two peaks at 71.0 and 73.0 eV, corresponding to $Pt^0$ and $Pt^{2+}$, respectively. The Fe 2p peak of Pd@FePt shifted left with an increase in the Pt amount, indicating that the electron loss from the Fe of $Fe_3O_4$ resulted in a binding energy increase of Fe 2p (Figure 3a). In contrast, the Pt 4f peak shifted right with an increase in the Fe amount, indicating a binding energy decrease of Pt 4f with electron gain (Figure 3b). We believe that the Fe of $Fe_3O_4$ donated electrons to Pt, which enhanced the electrocatalytic properties of Pd@FePt nanoparticles.

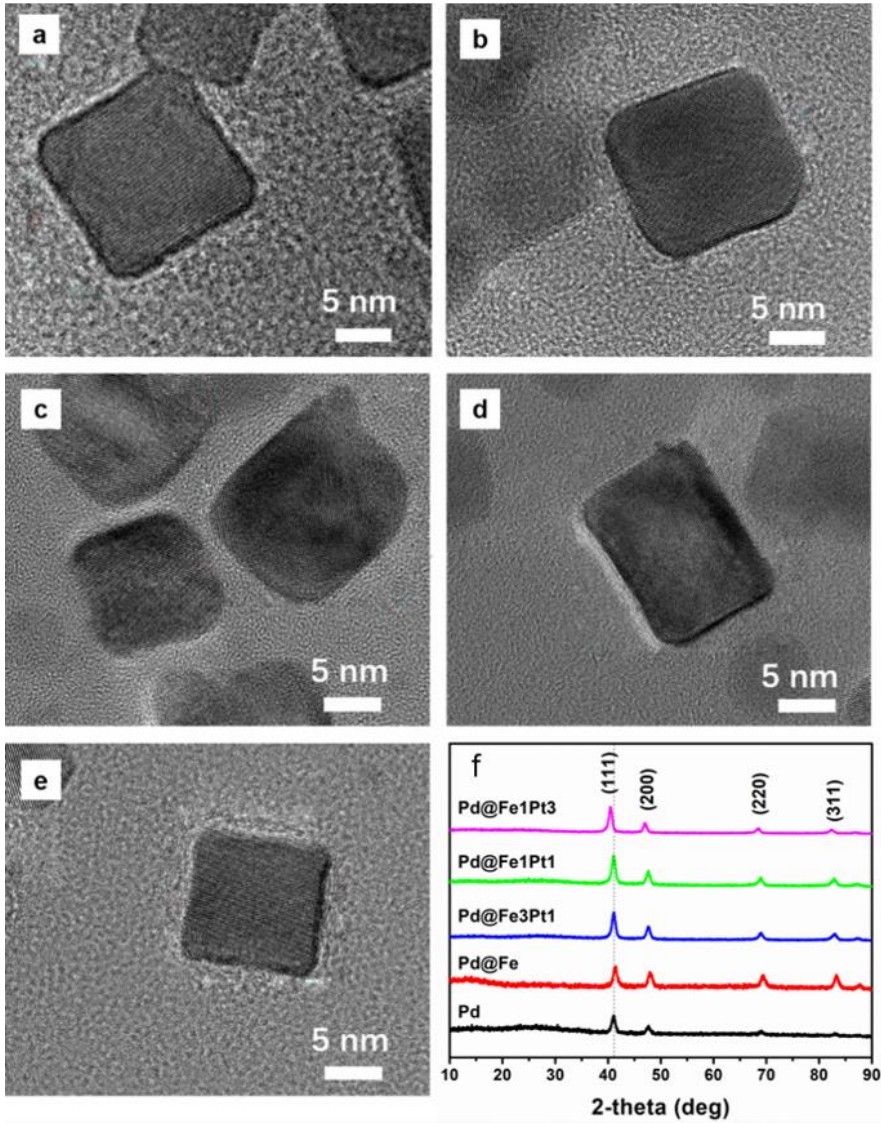

**Figure 1.** High-resolution TEM (HRTEM) images of (**a**) Pd, (**b**) Pd@Fe, (**c**) Pd@FePt$_{3/1}$, (**d**) Pd@FePt$_{1/1}$, and (**e**) Pd@FePt$_{1/3}$ core–shell nanoparticles. (**f**) XRD patterns of the nanoparticles.

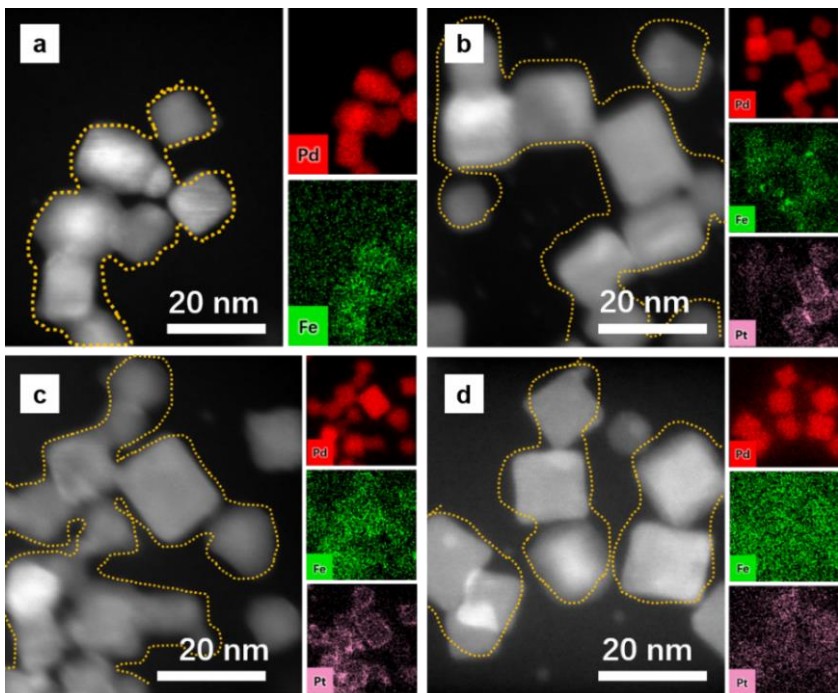

**Figure 2.** Energy-dispersive X-ray spectroscopy (EDS) mapping images of (**a**) Pd@Fe, (**b**) Pd@FePt$_{3/1}$, (**c**) Pd@FePt$_{1/1}$, and (**d**) Pd@FePt$_{1/3}$.

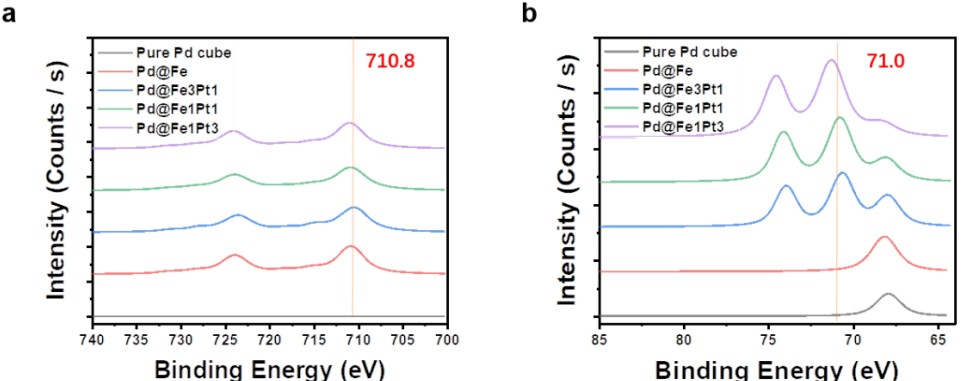

**Figure 3.** X-ray photoelectron spectroscopy (XPS) of (**a**) Fe 2p and (**b**) Pt 4f spectra of Pd, Pd@Fe, and Pd@FePt nanoparticles.

The synthesized Pd@Fe and Pd@FePt nanoparticles were supported on carbon particles to perform electrochemical reactions. The TEM images shown in Figure S5 confirmed that the Pd@Fe and Pd@FePt nanoparticles were deposited on carbon. This deposition was prepared by a simple dispersion process, and certain amounts of Pd@Fe or Pd@FePt nanoparticles were added to an ethanol solution involving carbon and stirred for 1 h at 73 °C. The morphologies and average diameters of Pd@Fe and Pd@FePt nanoparticles did not change after deposition on carbon.

The electrochemical catalytic activity of Pd@FePt catalysts for MOR was evaluated using cyclic voltammetry (CV). Figure 4 shows the CV curves of Pd@FePt catalysts in an electrolyte containing 1 M $CH_3OH$ and 1 M KOH. The current density is based on the loading amount of Pt. Pt is one of the most active substances in MOR [21]. However, as the Pt content of the Pd@FePt catalyst increased, the MOR current density did not proportionally increase and showed a maximum at a specific Pt content (Pd@FePt$_{1/1}$), indicating that the MOR activity improved because of the synergistic effect between the core–shell structure of the Pd@FePt catalyst and the electronic interaction between Fe and

Pt. It has been reported that both the Pd@Pt core–shell structure and the FePt alloy can increase MOR activity by appropriately controlling the binding energy of OH or the reaction intermediate of MOR through electronic interactions [14,22–27]. However, it is unclear which of the two effects dominate, but the synergy between the two effects was the greatest at PdFePt$_{1/1}$. Pd@FePt$_{1/1}$ showed higher catalytic activity despite its electrochemical surface area being smaller than that of commercial Pt/C, which can be observed in the CV curves based on the geometric current density (Figures S6 and S7). The electrochemical stability of Pd@FePt catalysts during MOR was also tested, as shown in Figure 5. The decrease in current density during the test is associated with catalyst degradation, which is mainly caused by CO species poisoning [26]. CO species are the reaction intermediates of MOR. However, the current densities of Pd@FePt catalysts remained higher than those of the commercial Pt/C throughout the test. In particular, the current density of Pd@FePt$_{1/1}$ was maintained at the highest, indicating that the stability of Pd@FePt$_{1/1}$ was the highest. This is also attributed to the aforementioned synergistic effect. The core–shell structure and electronic interaction between Pt and Fe reduce the binding strength of CO species and mitigate poisoning by CO species [22]. However, in the Pd@FePt catalysts, the current reduction rate for 1 h was greater than that of Pt/C. This is presumed to be because the relatively chemically unstable transition metal Fe is deformed during the stability test. Among the Pd@FePt catalysts, Pd@FePt$_{1/1}$ had the lowest current reduction rate, indicating that the stoichiometry also affects the stability of the catalyst.

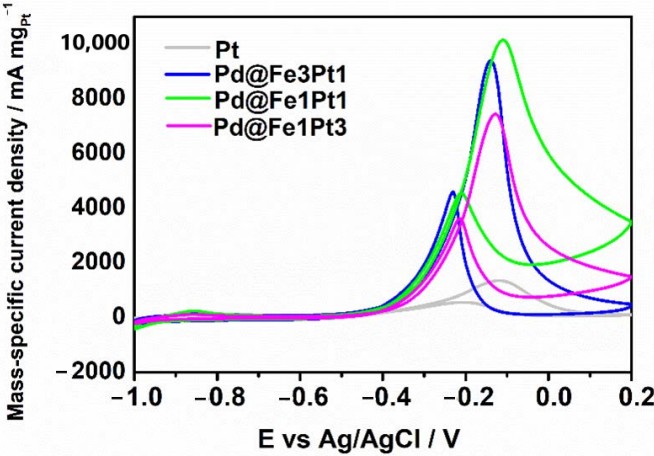

**Figure 4.** Cyclic voltammetry (CV) curves of Pd@FePt catalysts in 1 M CH$_3$OH + 1 M KOH electrolyte. The scan rate was 50 mV s$^{-1}$, and the current density was based on the loading amount of Pt.

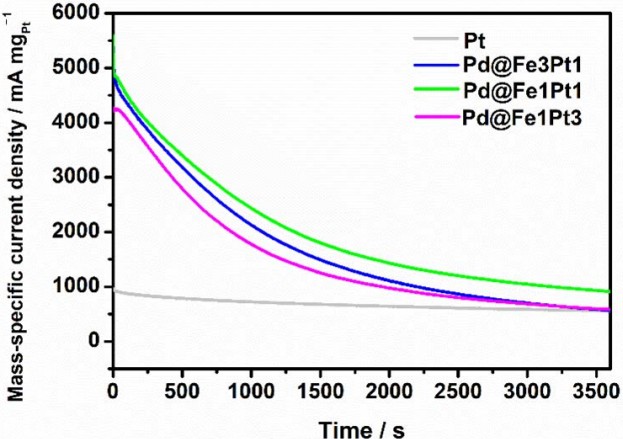

**Figure 5.** Current–time curves of Pd@FePt catalysts in 1 M CH$_3$OH + 1 M KOH electrolyte. The current density was recorded at −0.2 V for 3600 s and was based on the loading amount of Pt.

## 3. Materials and Methods

### 3.1. Materials

Polyvinylpyrrolidone (PVP, MW = 55,000), L-ascorbic acid (AA), sodium tetrahydridoborate ($NaBH_4$), potassium bromide (KBr), potassium tetrachloroplatinate ($K_2PtCl_4$, 99.99%), sodium tetrachloropalladate ($Na_2PdCl_4$, 98%), iron (III) chloride ($FeCl_3$), methanol ($CH_3OH$, ≥99.9%), potassium hydroxide pellet (KOH, ≥85%), and Nafion solution were purchased from Sigma–Aldrich. Carbon black (Vulcan XC-72) was purchased from CABOT (St. Louis, MO, USA). All chemicals were used without further purification.

### 3.2. Synthesis of Pd, Pd@Fe, and Pd@FePt Core–Shell Nanoparticles

Pd@Fe and Pd@FePt core–shell nanoparticles were synthesized using a direct seed-mediated growth method. PVP (36.93 mg), AA (50 mg), and KBr (300 mg) were dissolved in 8 mL deionized (DI) water, and the solution was heated to 80 °C while being stirred at 800 rpm. Then, 3 mL aqueous solution containing $Na_2PdCl_4$ (57 mg) was injected and heated to 80 °C while being stirred at 800 rpm for 3 h to prepare Pd cubes. After the solution was cooled to room temperature, 1 mL aqueous solution containing $FeCl_3$ (6.28 mg for Pd@Fe core–shell nanoparticles) or a mixture of $FeCl_3$ and $K_2PtCl_4$ with different Fe/Pt atomic ratios (4.71 mg and 4.02 mg for an Fe/Pt ratio of 3/1, 3.14 mg and 8.04 mg for an Fe/Pt ratio of 1/1, and 1.57 mg and 12.05 mg for an Fe/Pt ratio of 1/3) was added to the Pd dispersion and cooled to 0 °C. Then, 2 mL $NaBH_4$ solution (10 mg/mL) was added under vigorous stirring. After 0.5 h, the products were collected by centrifugation and washed three times with a DI water–acetone mixture.

### 3.3. Synthesis of Carbon-Supported Pd, Pd@Fe, and Pd@FePt Nanoparticles

The concentrations of Pd, Pd@Fe, and Pd@FePt nanoparticle suspensions were measured using ICP; then, a 3 mg metal catalyst was added into an ethanol solution involving 15 mg Vulcan XC-72 carbon (the metal/carbon compound was 20 wt.%). The final volume of the reacting solution was 20 mL. After being heated to 75 °C for 1 h under stirring at 800 rpm, carbon-supported nanoparticles were collected by centrifugation and redispersed in 1 mL ethanol. The metal concentration in the ethanol solution was kept at 3 mg/mL.

### 3.4. Characterization

TEM images were captured using a JEM-2100F microscope operated at 200 kV (JEOL, Akishima, Tokyo, Japan). HRTEM images were captured using an FEI Titan (Thermo Fisher Scientific, Waltham, MA, USA) microscope operated at 300 kV, and EDS mapping images were obtained using FEI Talos-F200X (Thermo Fisher Scientific, Waltham, MA, USA) at 200 kV with a dwell time of 290 μs/pixel. ICP analyses were performed using a direct reading echelle ICP spectrometer (Leeman, Hudson, NH, USA). XRD patterns were measured using a Rigaku D-MAX/A diffractometer at 35 kV and 35 mA (Rigaku, Akishima, Tokyo, Japan). XPS results were checked by K-Alpha (Thermo Electron) using a PHI 5000 VersaProbe (ULVAC PHI, Chigasaki, Kanagawa, Japan).

### 3.5. Electrochemical Characterization and Catalytic Activity Measurement

All electrochemical tests were conducted in a standard three-electrode cell linked to a CHI 760E potentiostat (CH Instruments, Austin, TX, USA) at room temperature. A Pt mesh ($1 \times 1$ cm$^{-2}$) and a leak-free AgCl/Ag/KCl electrode made up the counter electrode and reference electrode, respectively. A glassy carbon (GC) rotating disk electrode (RDE, 5 mm diameter, Pine Research Instrumentation, Durham, NC, USA) was used as the working electrode to build the three-electrode system for electrochemical measurements. The GC electrode was polished using $\alpha$-$Al_2O_3$ polishing powder and dried before the catalyst (20 wt.% catalyst metal on Vulcan XC-72) suspension loading. Of the prepared carbon-combined catalyst suspension, 4.7 μL was dropped onto the GC RDE with a geometric area of 0.196 cm$^2$. The total metal loading was 7.84 μg, and that for all catalysts was 40 μg cm$^{-2}$. After the link suspension solvent evaporated, the electrode was ready for

MOR measurement. The electrode surface was cleaned first by cycling between $-0.9$ and 0.3 V in 1 M aqueous KOH (20–40 cycles); consequently, the MOR process was performed. The electrochemical measurements for the MOR were carried out at room temperature in a solution containing 1 M $CH_3OH$ and 1 M KOH under a flow of $N_2$. CV curves were collected at a sweep rate of 50 mV/s. A test of chronopotentiometry (CA) was performed for 3600 s by polarizing at 0.2 V.

## 4. Conclusions

In summary, we have developed a simple synthetic route for Pd@FePt core–shell nanoparticles using a direct seed-growth method. Owing to the use of a strong reducing agent, a transition metal such as Fe could be incorporated into the Pt shell, leading to the FePt shell formation on the surface of Pd seeds. The synthesized Pd@FePt nanoparticles exhibited FePt shell-dependent catalytic properties, and Pd@FePt$_{1/1}$ showed the highest catalytic activity and durability toward MOR. The synergistic effect and electronic interaction between Fe and Pt play critical roles in catalytic activity improvement. We believe that multi-metallic nanoparticles via the direct seed-mediated growth method could be used in many applications, including electrocatalysis.

**Supplementary Materials:** The following are available online at https://www.mdpi.com/2073-4344/11/1/130/s1. Figure S1: TEM images and size distribution of Pd@Fe and Pd@FePt nanoparticles. Figure S2: HAADF-STEM-EDS mapping images and EDS line scanning profiles of Pd@Fe and Pd@FePt nanoparticles. Figure S3: TEM images of Pd@FePt$_{1/3}$ using AA instead of NaBH$_4$. Figure S4: Fe 2p$_{3/2}$ XPS peak differentiation-imitating analysis of Pd@Fe and Pd@FePt nanoparticles. Figure S5: retention of ECSA in an alkaline media (a) and their value (b). Figure S6: TEM images of Pd@Fe/C and Pd@FePt/C. Table S1: ICP results of Pd@Fe and Pd@FePt nanoparticles. Table S2: ICP results of Pd@Fe/C and Pd@FePt/C.

**Author Contributions:** Conceptualization, synthesis, and writing—original draft preparation, X.X. and E.J.; conceptualization, electrocatalysts, writing—original draft preparation, S.Y.; characterization, H.K.; characterization, H.-K.K.; writing—review and editing, K.-Y.L.; supervision, methodology, further data analysis, and writing—review and editing, J.-P.A., T.L., J.K., and T.Y. All authors have read and agreed to the published version of the manuscript.

**Funding:** This work was supported by the NRF grant funded by the Korean government (MSIP) (NRF-2014R1A5A1009799, NRF-2016M3D1A1021140, NRF-2017R1A5A1015365, NRF-2020R1A2C1003885, NRF-2019R1A6C1010052, and NRF-2020M3A7B4002030).

**Conflicts of Interest:** The authors declare no conflict of interest.

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
