# Peer review of "Facile Aqueous–Phase Synthesis of Pd–FePt Core–Shell Nanoparticles for Methanol Oxidation Reaction"

_catalysts, doi:10.3390/catal11010130_

Round 1
Reviewer 1 Report
Please see the attached document for my review comments.

Author Response
Taekyung Tu, Ph.D.
Associate Professor
Department of Chemical Engineering
Kyung Hee University
Reviewer#1
Comment 1. Abstract
(1) The statement “making them deployable in direct methanol fuel cells (DMFC)” is rather strong and I doubt that this conclusion can be made just out of the MOR measurement as more properties are needed for a proper DMFC material. However, it does look promising, so maybe just soften the sentence a bit.
Response 1-1. We appreciate the reviewer’s thoughtful comment. As you mentioned, we changed the sentence from “making them deployable in direct methanol fuel cells (DMFC)” to “making them efficient in direct methanol fuel cells (DMFC)”.
(Please refer lines 27–28 in page 1.)
Comment 2. Results
(1) I suggest to split the very first sentence (line 65-67) of the results section after “…Pd nanocubes as seeds” for better readability.
Response 2-1. To make readable manuscript, we revised the first sentence of the results section.
“Pd@Fe and Pd@FePt core–shell nanoparticles were synthesized by a direct seed-mediated method in an aqueous solution. Fe and a mixture of Fe and Pt precursors for shell formation were used with NaBH4 as a reducing agent on Pd nanocubes as seeds.”
(Please refer lines 63–65 in page 2.)
(2) In line 70, make it clear that the ratio is a molecular weight ratio (Fe vs Pt).
Response 2-2. As reviewer’s advice, we exactly designated a molecular weight ratio of Fe and Pt in the manuscript.
“When we synthesized Pd@FePt nanoparticles, we used different molecular weight ratios of Fe and Pt precursors, that is, 3/1, 1/1, and 1/3, and nominally named them as Pd@FePt3/1, Pd@FePt1/1, and Pd@FePt1/3, respectively.”
(Please refer lines 67–69 in page 2.)
(3) The mentioned “20 %” of atomic shell content seems to be rather 15-19 % looking at Table S1 as suggested (line 73-74). Please revise.
Response 2-3. We modified the value of the atomic shell content from “20 %” to “15-19%” in the manuscript.
(Please refer lines 70 in page 2.)
(4) The sentence in lines 111-113 about the presence of Fe3O4 structure referring to Figure S2 is really important as it supports the already mentioned claim. However, as it is written, it does not link back to this statement. Please revise.
Response 2-4. As reviewer pointed out, to clearly connect between Figures S2 and 3, we correctly revised electron source from Fe of Fe3O4 in the manuscript.
“The Fe 2p peak of Pd@FePt shifted left with an increase in the Pt amount, indicating that the electron loss from Fe of Fe3O4 resulted in the binding energy increase of Fe 2p (Figure 3a). In contrast, the Pt 4f peak shifted right with an increase in the Fe amount, indicating the binding energy decrease of Pt 4f with electron gain (Figure 3b). We believe that Fe of Fe3O4 donated electrons to Pt, which enhanced the electrocatalytic properties of Pd@FePt nanoparticles.”
(Please refer lines 109-113 in page 4.)
(5) In lines 82-86, the differences in XRD peak angles are discussed (compare Figure 1 f)). What about the difference in height/intensity though? How can their change be explained?
Response 2-5. In general, XRD intensity have a correlation with relative crystallinity of samples, especially related to size. We can explain that Pd nanoparticles have a lower intensity comparted to Pd@Fe and Pd@FePt nanoparticles because of smaller size of them. However, we thought that it is not critical result to write in this research. In addition, there were not big differences between all samples (Pd, Pd@Fe, and Pd@FePt nanoparticles). It is a reason that we focused the shifts in XRD peak angles.
(6) If possible, add a reference to “Pt is one of the most active substances in MOR” (line 131).
Response 2-6. We added a reference related to Pt as excellent electro-catalysts for MOR.
Please check ref. 21. (J. Mater. Chem. A 2018, 6, 18165-18172.)
(7) The claim that Pd@FePt is stable over time is not very strongly supported by a 1h measurement, where the current density drops by about 80 %. It is true and therefore still promising, that it is higher compared to pure Pt, but there is already an overlap for some of the samples after 3600 s. Please soften/revise this part.
Response 2-7. We agree that the current density reduction was greater in our samples than Pt/C and some of our samples showed almost the same current density with that of Pt/C at 3,600. We have added the comments on this to the manuscript.
“However, in the Pd@PtFe catalysts, the current reduction rate for 1 hour was greater than that of Pt/C. This is presumed to be because the relatively chemically unstable transition metal Fe is deformed during the stability test. Among the Pd@FePt catalysts, Pd@FePt1/1 had the lowest current reduction rate, indicating that the stoichiometry also affects the stability of the catalyst.”
(Please refer line 140-151 in page 4)
Comment 3. Materials and Methods
(1) I think that “Fe/Pd” in line 176 should be “Fe/Pt”. Please revise if applicable.
Response 3-1. Due to reviewer’s comment, we could correct our mistake. We changed from “Fe/Pd” to “Fe/Pt”.
(Please refer line 167 in page 6)
(2) In 3.2., the process for the different Fe/Pt ratios is explained. However, I am unable to follow how the values in the text as well as the ones in Table S1 lead to ratios of 3/1, 1/1 and 1/3, assuming molecular weights of ~56 u and ~195 u for Fe and Pt, respectively. Please explain.
Response 3-2. As we mentioned in the manuscript, we denoted Pd@FePt nanoparticles depending on molecular weight ratio of Fe and Pt precursors when we injected to synthesize. To avoid confusion, we added the word of ‘nominally’ in the sentence.
“When we synthesized Pd@FePt nanoparticles, we used different molecular weight ratios of Fe and Pt precursors, that is, 3/1, 1/1, and 1/3, and nominally named them as Pd@FePt3/1, Pd@FePt1/1, and Pd@FePt1/3, respectively.”
(Please refer lines 67–69 in page 2.)
In addition, many papers have followed this way to name their samples, corresponding the references.
- a) Am. Chem. Soc. 2020, 142, 14688-14701.
- b) Cat. B: Environ. 2019, 252, 10-17.
- c) Cat. B: Environ. 2018, 234, 10-18.
(3) In 3.5., you might want to change the word “slurry” to e.g. “suspension” as “slurry” usually appears in the context of manure.
Response 3-3. According to the reviewer’s suggestion, we changed the word from “slurry” to “suspension” in the manuscript.
“The GC electrode was polished using α-Al2O3 polishing powder and dried before the catalyst (20 wt.% catalyst metal on Vulcan XC-72) suspension loading. Of the prepared carbon-combined catalyst suspension, 4.7 μL was dropped onto the GC RDE with a geometric area of 0.196 cm2. The total metal loading was 7.84 μg, and that for all catalysts was 40 μg cm−2. After the link suspension solvent evaporated, the electrode was ready for MOR measurement.”
(Please refer line 194-198 in page 4)
Comment 4. Figures
(1) If possible and where applicable, use the same color code for all graphs like 1 f), 3-5, S1 e+f, S6+7 (e.g. black for pure Pd, red for Pd-Fe…).
(2) In S1 f), there is no need for the y-axis to start at 0 with a break at 4 nm. It can also just start at a higher value and span from e.g. 10-20 nm. Moreover, it should not be a scatter-line plot, but rather a scatter plot, as the property measured comes from different materials. The same labelling as in S1 e) would also help the comparison between both figures.
Response 4-1 and 4-2. We appreciate for the comments. We revised colors in figures 1f, 3-5, S1 e+f, S6, and S7 for each samples (Pd, Pd@Fe, Pd@FePt nanoparticles). Moreover, in Figure S1 f, the y-axis starts at 10 nm.
(3) Why are there two scans in Figure 4 and S7. It is not mentioned anywhere in the text. Please explain this for better readability.
Response 4-3. Thanks for the comment. Figures 4 and S7 are CV graphs in which the current was normalized based on the Pt loading amount and the geometrical area of the electrode, respectively. We mentioned this in the manuscript and in the figure captions of Figures 5 and S7. Please check lines 122-132 in page 4 of the manuscript, and the figure captions of Figures 5 and S7.
Comment 5. Tables
(1) Table S2 should be on one page and not spanning over two pages. Please revise.
Response 5-1. We reduce the size of Table S2 to be not over two pages.
Comment 6. References
(1) The references are not numbered (at all) as they appear in the manuscript. Please revise.
Response 6-1. We checked the references in the manuscript, however, all references were numbered. Please, check again.
Comment 7. Supplementary
(1) The additional information in the supplementary material document is really helpful and of high quality. However, I suggest to add some descriptive text to each item. This would not only help the readability/ back-linking to the corresponding parts in the manuscript, but also improve the quality in general of the supplementary section as a stand-alone document.
Response 7-1. We would like to thank you for making better manuscript. After reviewer’s suggestion, we thought how to add more comments in the manuscript for supplementary. In conclusion, we believe that the revised manuscript is the best version to explain our results.

Reviewer 2 Report
The presented work demonstrates through the performed characterizations that, its can be obtained Pd@Fe and Pd@FePt core–shell nanoparticles with excellent activity and stability toward methanol oxidation reaction, by simple synthetic route using a direct seed-growth method.
The results obtined are complex, well presented. The bibliographic resources are up to date.
Therefore, I believe that the work can be published.
Author Response
Taekyung Tu, Ph.D.
Associate Professor
Department of Chemical Engineering
Kyung Hee University
January 07, 2021
Reviewer#1
Comment 1. Abstract
(1) The statement “making them deployable in direct methanol fuel cells (DMFC)” is rather strong and I doubt that this conclusion can be made just out of the MOR measurement as more properties are needed for a proper DMFC material. However, it does look promising, so maybe just soften the sentence a bit.
Response 1-1. We appreciate the reviewer’s thoughtful comment. As you mentioned, we changed the sentence from “making them deployable in direct methanol fuel cells (DMFC)” to “making them efficient in direct methanol fuel cells (DMFC)”.
(Please refer lines 27–28 in page 1.)
Comment 2. Results
(1) I suggest to split the very first sentence (line 65-67) of the results section after “…Pd nanocubes as seeds” for better readability.
Response 2-1. To make readable manuscript, we revised the first sentence of the results section.
“Pd@Fe and Pd@FePt core–shell nanoparticles were synthesized by a direct seed-mediated method in an aqueous solution. Fe and a mixture of Fe and Pt precursors for shell formation were used with NaBH4 as a reducing agent on Pd nanocubes as seeds.”
(Please refer lines 63–65 in page 2.)
(2) In line 70, make it clear that the ratio is a molecular weight ratio (Fe vs Pt).
Response 2-2. As reviewer’s advice, we exactly designated a molecular weight ratio of Fe and Pt in the manuscript.
“When we synthesized Pd@FePt nanoparticles, we used different molecular weight ratios of Fe and Pt precursors, that is, 3/1, 1/1, and 1/3, and nominally named them as Pd@FePt3/1, Pd@FePt1/1, and Pd@FePt1/3, respectively.”
(Please refer lines 67–69 in page 2.)
(3) The mentioned “20 %” of atomic shell content seems to be rather 15-19 % looking at Table S1 as suggested (line 73-74). Please revise.
Response 2-3. We modified the value of the atomic shell content from “20 %” to “15-19%” in the manuscript.
(Please refer lines 70 in page 2.)
(4) The sentence in lines 111-113 about the presence of Fe3O4 structure referring to Figure S2 is really important as it supports the already mentioned claim. However, as it is written, it does not link back to this statement. Please revise.
Response 2-4. As reviewer pointed out, to clearly connect between Figures S2 and 3, we correctly revised electron source from Fe of Fe3O4 in the manuscript.
“The Fe 2p peak of Pd@FePt shifted left with an increase in the Pt amount, indicating that the electron loss from Fe of Fe3O4 resulted in the binding energy increase of Fe 2p (Figure 3a). In contrast, the Pt 4f peak shifted right with an increase in the Fe amount, indicating the binding energy decrease of Pt 4f with electron gain (Figure 3b). We believe that Fe of Fe3O4 donated electrons to Pt, which enhanced the electrocatalytic properties of Pd@FePt nanoparticles.”
(Please refer lines 109-113 in page 4.)
(5) In lines 82-86, the differences in XRD peak angles are discussed (compare Figure 1 f)). What about the difference in height/intensity though? How can their change be explained?
Response 2-5. In general, XRD intensity have a correlation with relative crystallinity of samples, especially related to size. We can explain that Pd nanoparticles have a lower intensity comparted to Pd@Fe and Pd@FePt nanoparticles because of smaller size of them. However, we thought that it is not critical result to write in this research. In addition, there were not big differences between all samples (Pd, Pd@Fe, and Pd@FePt nanoparticles). It is a reason that we focused the shifts in XRD peak angles.
(6) If possible, add a reference to “Pt is one of the most active substances in MOR” (line 131).
Response 2-6. We added a reference related to Pt as excellent electro-catalysts for MOR.
Please check ref. 21. (J. Mater. Chem. A 2018, 6, 18165-18172.)
(7) The claim that Pd@FePt is stable over time is not very strongly supported by a 1h measurement, where the current density drops by about 80 %. It is true and therefore still promising, that it is higher compared to pure Pt, but there is already an overlap for some of the samples after 3600 s. Please soften/revise this part.
Response 2-7. We agree that the current density reduction was greater in our samples than Pt/C and some of our samples showed almost the same current density with that of Pt/C at 3,600. We have added the comments on this to the manuscript.
“However, in the Pd@PtFe catalysts, the current reduction rate for 1 hour was greater than that of Pt/C. This is presumed to be because the relatively chemically unstable transition metal Fe is deformed during the stability test. Among the Pd@FePt catalysts, Pd@FePt1/1 had the lowest current reduction rate, indicating that the stoichiometry also affects the stability of the catalyst.”
(Please refer line 140-151 in page 4)
Comment 3. Materials and Methods
(1) I think that “Fe/Pd” in line 176 should be “Fe/Pt”. Please revise if applicable.
Response 3-1. Due to reviewer’s comment, we could correct our mistake. We changed from “Fe/Pd” to “Fe/Pt”.
(Please refer line 167 in page 6)
(2) In 3.2., the process for the different Fe/Pt ratios is explained. However, I am unable to follow how the values in the text as well as the ones in Table S1 lead to ratios of 3/1, 1/1 and 1/3, assuming molecular weights of ~56 u and ~195 u for Fe and Pt, respectively. Please explain.
Response 3-2. As we mentioned in the manuscript, we denoted Pd@FePt nanoparticles depending on molecular weight ratio of Fe and Pt precursors when we injected to synthesize. To avoid confusion, we added the word of ‘nominally’ in the sentence.
“When we synthesized Pd@FePt nanoparticles, we used different molecular weight ratios of Fe and Pt precursors, that is, 3/1, 1/1, and 1/3, and nominally named them as Pd@FePt3/1, Pd@FePt1/1, and Pd@FePt1/3, respectively.”
(Please refer lines 67–69 in page 2.)
In addition, many papers have followed this way to name their samples, corresponding the references.
- a) Am. Chem. Soc. 2020, 142, 14688-14701.
- b) Cat. B: Environ. 2019, 252, 10-17.
- c) Cat. B: Environ. 2018, 234, 10-18.
(3) In 3.5., you might want to change the word “slurry” to e.g. “suspension” as “slurry” usually appears in the context of manure.
Response 3-3. According to the reviewer’s suggestion, we changed the word from “slurry” to “suspension” in the manuscript.
“The GC electrode was polished using α-Al2O3 polishing powder and dried before the catalyst (20 wt.% catalyst metal on Vulcan XC-72) suspension loading. Of the prepared carbon-combined catalyst suspension, 4.7 μL was dropped onto the GC RDE with a geometric area of 0.196 cm2. The total metal loading was 7.84 μg, and that for all catalysts was 40 μg cm−2. After the link suspension solvent evaporated, the electrode was ready for MOR measurement.”
(Please refer line 194-198 in page 4)
Comment 4. Figures
(1) If possible and where applicable, use the same color code for all graphs like 1 f), 3-5, S1 e+f, S6+7 (e.g. black for pure Pd, red for Pd-Fe…).
(2) In S1 f), there is no need for the y-axis to start at 0 with a break at 4 nm. It can also just start at a higher value and span from e.g. 10-20 nm. Moreover, it should not be a scatter-line plot, but rather a scatter plot, as the property measured comes from different materials. The same labelling as in S1 e) would also help the comparison between both figures.
Response 4-1 and 4-2. We appreciate for the comments. We revised colors in figures 1f, 3-5, S1 e+f, S6, and S7 for each samples (Pd, Pd@Fe, Pd@FePt nanoparticles). Moreover, in Figure S1 f, the y-axis starts at 10 nm.
(3) Why are there two scans in Figure 4 and S7. It is not mentioned anywhere in the text. Please explain this for better readability.
Response 4-3. Thanks for the comment. Figures 4 and S7 are CV graphs in which the current was normalized based on the Pt loading amount and the geometrical area of the electrode, respectively. We mentioned this in the manuscript and in the figure captions of Figures 5 and S7. Please check lines 122-132 in page 4 of the manuscript, and the figure captions of Figures 5 and S7.
Comment 5. Tables
(1) Table S2 should be on one page and not spanning over two pages. Please revise.
Response 5-1. We reduce the size of Table S2 to be not over two pages.
Comment 6. References
(1) The references are not numbered (at all) as they appear in the manuscript. Please revise.
Response 6-1. We checked the references in the manuscript, however, all references were numbered. Please, check again.
Comment 7. Supplementary
(1) The additional information in the supplementary material document is really helpful and of high quality. However, I suggest to add some descriptive text to each item. This would not only help the readability/ back-linking to the corresponding parts in the manuscript, but also improve the quality in general of the supplementary section as a stand-alone document.
Response 7-1. We would like to thank you for making better manuscript. After reviewer’s suggestion, we thought how to add more comments in the manuscript for supplementary. In conclusion, we believe that the revised manuscript is the best version to explain our results.
